# Clinical Manifestation of Cytomegalovirus-Associated Protein-Losing Enteropathy in Children

**DOI:** 10.3390/nu15132844

**Published:** 2023-06-22

**Authors:** Claire Ferrua, Anais Lemoine, Alexis Mosca, Anne-Aurélie Lopes

**Affiliations:** 1Paediatric Emergency Department, AP-HP, Robert Debré Hospital, Université Paris Cité, 48 Boulevard Sérurier, 75019 Paris, France; 2Paediatric Nutrition and Gastroenterology Department, AP-HP, Trousseau Hospital, Sorbonne Université, 26 Avenue du Dr Arnold Netter, 75012 Paris, France; 3Paediatric Gastroenterology Department, AP-HP, Robert Debré Hospital, Université Paris Cité, 48 Boulevard Sérurier, 75019 Paris, France

**Keywords:** protein-losing enteropathy, cytomegalovirus, generalised oedema, endoscopy, immunosuppressed

## Abstract

In children, CMV-associated protein-losing enteropathy (PLE) is characterised by a benign course and spontaneous healing but can lead to generalised oedema. Poorly defined, it is diagnosed after unnecessary invasive tests. Children with CMV-associated PLE between 2009 and 2019 in two French hospitals are retrospectively described. Clinical and biological signs, CMV identification, endoscopy and histological findings, disease management and course are analysed. CMV-associated PLE is proven in 21 immunocompetent and 22 immunosuppressed patients, with ages consistent with primo-infection and reactivation, respectively. The digestive symptoms prevail in immunocompetent children, mainly with vomiting (85.7% versus 50%, CI [1.2; 39.2], *p* = 0.02). Immunocompetent patients show more oedema (61.9% versus 4.5%, CI [3.6; 1502.4], *p* < 0.001), linked to more severe hypoalbuminemia (21.2 g/L [17.6–25.7] versus 29.6 g/L [24.9–33.9], *p* = 0.01). A severe course is observed in 23.8% of the immunocompetent patients and 54.5% of the immunosuppressed ones (*p* = 0.06). Evidence of CMV infection based on non-invasive methods is found on 88.9% of immunocompetent and 95.5% of immunosuppressed patients (*p* = 0.58), while endoscopy was performed on 95.2% and 100% of them, respectively (*p* = 0.48), without any therapeutic change. Thus, CMV-associated PLE should be suspected in children with generalised oedema. Not as benign as previously described, it can be confirmed using non-invasive tests.

## 1. Introduction

Cytomegalovirus (CMV) is a widespread virus, transmitted through any body fluids, with a prevalence ranging from 45 to 100%, depending on geographic areas and socioeconomic status [1]. Infants aged six months to two years have the highest rates of CMV expression, with more than 50% of children in day-care centres having positive CMV excretion [2]. Mostly unnoticed in healthy people [3,4,5,6], CMV infections can be severe in immunosuppressed patients with a large range of manifestations such as hepatitis, retinitis, pneumonia or neurological symptoms [6,7]. Gastrointestinal (GI) involvement is also well documented in immunosuppressed patients but little in immunocompetent ones [8]. The most typical GI manifestation is vomiting followed within 15 days by oedema due to CMV-associated protein-losing enteropathy (PLE) [9]. Known as Menetrier’s disease in adults and rarely described in the paediatric literature, it is characterised by an acute onset, a benign course and spontaneous healing when it occurs in immunocompetent children [10], whereas antiviral treatment is required for immunosuppressed patients [6,7].

However, faced with generalised oedema in paediatric emergency departments, the physician will consider cardiac, renal or hepatic causes, which require specific and urgent treatments, and will perform the first inevitable biological tests in accordance, without considering a CMV infection [11]. In CMV-associated PLE, the biological tests will reveal hypoproteinaemia and hypoalbuminemia, such as for some renal causes, but without urinary test abnormalities [12,13,14]. However, as CMV-associated PLE is not a common diagnosis, unnecessary tests will then be performed to try to understand the cause of oedema. Finally, the diagnosis is, in most cases, established by a gastroenterologist, after an endoscopy [15,16], whereas a medical history of digestive symptoms days before oedema can guide the physician toward CMV-associated PLE and lead to confirm the diagnosis through non-invasive methods such as serology or CMV DNA detection using polymerase chain reaction (PCR) [4,6].

In this study, we describe the clinical, biological and histological characteristics of the largest cohort of immunocompetent and immunosuppressed children with CMV-associated PLE.

## 2. Materials and Methods

A retrospective descriptive study of children up to 18 years of age with CMV-associated PLE diagnosed between 2009 and 2019 was conducted in the Robert Debré and Trousseau hospitals, Paris, France. The list of patients who had biological evidence of CMV infection was obtained from the virology laboratories of each hospital complemented by the list of patients diagnosed with CMV-associated PLE from the electronic records.

The CMV infection was confirmed by at least one positive test, such as pp65 antigenemia, serological tests or PCR CMV performed on body fluids or digestive tissue. On serological tests, acute CMV infection was defined by the detection of CMV IgM antibodies or a fourfold increase in the level of CMV IgG specific antibodies about two to four weeks later. The patients were divided into two groups:Immunocompetent patients: patients who did not have an underlying disease or treatment which results in immunodeficiency.Immunosuppressed patients divided in 2 sub-groups:
Patients who have primary immunodeficiency diseases;Patients who have a secondary immunodeficiency state due to either haematologic disease, GI condition or immunosuppressive therapies in the context of organ or bone marrow transplant or inflammatory bowel disease (IBD).

To allow for a detailed description, clinical, biological, endoscopic and histological findings were retrieved. The history and the average time of onset of digestive symptoms before the first consultation and the clinical signs of PLE were recorded. Signs of heart failure (tender hepatomegaly, hepatojugular reflux and cardiomegaly on chest X-ray), renal failure (oliguria, hypertension, haematuria and proteinuria) and hepatic failure (hepatomegaly, splenomegaly and collateral circulation) were investigated to rule them out.

Hypoalbuminemia was defined by serum albumin <35 g/L. Hyponatremia was defined by a natremia ≤135 mmol/L. An inflammatory syndrome was defined by a CRP > 10 mg/L. The normal value of alpha-1-antitrypsin clearance was defined by <20 mL/day.

On the esophagogastroduodenoscopy (EGD) and colonoscopy, macroscopic description and location of lesions were pointed out [15]. The appearance of the mucosa at histopathological examination was graded as follows [17].

Normal;Non-specific aspect: signs of inflammation such as oedema of the chorion, ulceration, polynuclear infiltration, mononuclear cells infiltration, lymphoid follicles and villous atrophy;Evocative aspect of CMV: glandular abnormalities such as apoptosis, necrosis or crypt abscess or nuclear dystrophy;Typical feature of CMV: inclusions on histology with positive immunohistochemistry for CMV (antibodies against CMV).

Concerning management of CMV-associated PLE, the emergency treatment as therapies prescribed during hospitalisation were recorded. Regarding immunosuppressed patients, ongoing treatments were noted.

The disease course was assessed and classified into three stages according to the management and the evolution of the disease:Benign course: a benign initial feature of the disease, with a good general condition, good hydration state and prescription of symptomatic treatment or a short hospitalisation without any treatment except hydration;Mild course: a severe initial clinical presentation (poor general condition, dehydration, need of blood transfusion) with subsequent good evolution and recovery;Severe course: an overall serious illness with the presence of one of the following criteria: hospitalisation in an intensive care unit, hypovolemic chock, parenteral nutrition or death.

### Statistical Analyses

Data were expressed as median with interquartile range or minimum and maximum for quantitative variables or as numbers and percentages for qualitative variables with 95% confidence intervals (CI). Continuous variables were compared via the Mann–Whitney test. Qualitative variables were compared using the Chi² test or Fisher’s exact test, according to the number of cases. SPSS for Windows was used for all statistical calculations. The statistical significance was defined as *p* < 0.05.

## 3. Results

Between 2007 and 2019, 43 patients were diagnosed with documented CMV-associated PLE: 21 immunocompetent patients (48.8%) and 22 immunosuppressed patients (51.2%) (Figure 1). The median patient age at diagnosis was significantly younger for immunocompetent patients (29.7 months [1–133.7] versus 105.3 months [3.4–179.4], *p* < 0.001), with significantly more immunocompetent patients aged under 24 months (CI [1.2; 78.8], *p* = 0.02) (Table 1).

### 3.1. Clinical and Biological Signs

The median time between onset of symptoms and hypoalbuminemia was 9.5 days [6.3–18] in the immunocompetent group and 14 days [2–18] in the immunosuppressed one (*p* = 0.88) (Table 2). For both immunocompetent and immunosuppressed populations, the most frequent symptoms were vomiting, prevailing in immunocompetent patients (85.7% versus 50%, CI [1.2; 39.2], *p* = 0.02), acute diarrhoea (66.7% and 86.4%, CI [0.05; 1.7], *p* = 0.16), anorexia (66.7% and 68.2%, CI [0.2; 4], *p* = 1), asthenia (52.4% and 50%, CI [0.3; 4.3], *p* = 1) and fever (47.6% and 54.5%, CI [0.2; 2.9], *p* = 0.76). Oedema was mainly observed in immunocompetent children (61.9% versus 4.5%, CI [3.6; 1502.4], *p* < 0.001) with no signs of cardiac, renal or hepatic disease. Hypoalbuminemia was the most frequent biological sign for both groups and was noticed in 85.7% of immunocompetent children and in 95.5% of immunosuppressed ones (CI [0.005; 4], *p* = 0.34). The hypoalbuminemia was significantly more important in immunocompetent patients (21.2 g/L [17.6–25.7] versus 29.6 g/L [24.9–33.9], *p* = 0.01). Dehydration occurred in 42.8% of immunocompetent children versus 27.3% of immunosuppressed ones (CI [0.5; 8.8], *p* = 0.35). Hyponatremia was significantly more frequent in the immunocompetent group (57.1% versus 22.7%, CI [1; 21.4], *p* = 0.03). Alpha-1-antitrypsin clearance was never performed.

### 3.2. CMV Identification

All patients in the study had CMV detected by at least one viral test such as serology, pp65 antigenemia or PCR on bodily fluid (blood, urine) or digestive tissue (Table 3). Without regard to the result of CMV PCR on digestive tissue, CMV was detected in 16 immunocompetent patients (76.2%) and 21 immunosuppressed ones (95.5%) (CI [0.006; 8.1], *p* = 0.58), knowing that no other test than PCR on digestive tissue was performed on 3 immunocompetent patients. An endoscopy was performed on 20 immunocompetent patients (95.2%) and all immunosuppressed patients (*p* = 0.48). Among patients who had a CMV PCR performed on digestive tissue, the PCR was positive for 75% of the immunocompetent children and all immunosuppressed children (*p* = 0.02).

Regarding the endoscopic results of the immunocompetent patients, twelve patients had clinical signs of gastritis with seven positive gastric CMV PCR on the eleven biopsies performed, one patient had signs of oesophagitis with a positive CMV PCR on oesophageal biopsy, and eight patients had signs of colitis with seven positive CMV PCR on colic biopsies. Among the immunosuppressed patients, five had signs of gastritis, one of duodenitis and sixteen of colitis, all with positive CMV PCR on the matched biopsies. In the immunocompetent patients, EGD more often showed fundic gastritis (66.7% versus 20%, CI [1.3; 57.6], *p* = 0.01) and antral gastritis (61.1%% versus 40%, CI [0.5; 12], *p* = 0.3) with gastric hypertrophic folds (CI [1.5; +∞ [, *p* = 0.009) (Table A1). The presence of nuclear dystrophy was more observed in immunocompetent patients (57.9% versus 23.8%, CI [1; 21.6], *p* = 0.05), and typical features of CMV were observed in 31.6% and 27.3% of patients, respectively (*p* = 1).

### 3.3. Medical Management and Clinical Course

Rehydration was administered for 52.4% of immunocompetent and 54.5% of immunosuppressed patients (CI [0.2; 3.6], *p* = 1) (Table 4). All immunosuppressed patients and five immunocompetent patients were treated with a curative dose of the antiviral drug (100% versus 23.8%, *p* < 0.001). The other supportive treatments administered did not show a significant difference between both groups, except intravenous steroids (9.5% versus 40.9% CI [0.01; 0.9], *p* = 0.03).

The clinical course was globally different between the two groups (*p* = 0.02) with a benign course in 47.6% of the immunocompetent patients versus 18.2% of the immunosuppressed patients (CI [0.9; 21.7], *p* = 0.05) and a severe clinical course 23.8% of the immunocompetent patients versus 54.5% of the immunosuppressed ones (CI [0.06; 1.1], *p* = 0.06).

## 4. Discussion

To our best knowledge, this study is the largest cohort describing CMV-associated PLE in children. PLE is commonly considered in patients with chronic diarrhoea and peripheral oedema and can be due to many GI diseases, such as an allergy to cow’s milk protein in the youngest patients, primary intestinal lymphangiectasia, Henoch–Schoenlein purpura or syndromic diarrhoea [18,19,20,21,22]. However, CMV-associated PLE is poorly considered. In our study, the most commonly presenting symptoms of CMV-associated PLE were vomiting followed by generalised oedema associated with unspecific symptoms suggesting a viral illness, similar to the few case reports and short case series depicting this disease in children [12,13,14,23,24,25,26,27]. To characterise all the manifestations of CMV-associated PLE, we considered the immunological status of patients and did not just focus on gastropathy, as is often the case [12,13,14,23,24,25,26]. Digestive symptoms prevailed in immunocompetent patients, and an altered general state was observed in both groups. In the immunosuppressed patients, the underlying disease could be a confounding factor for asthenia and anorexia, but their presence in immunocompetent patients, often associated with clinical signs of dehydration and hyponatremia, supports the role of CMV infection in this altered state. Oedema also prevailed in immunocompetent patients (61.9%). Oedema is linked to hypoalbuminemia resulting from protein leakage through the GI mucosa and appeared approximately ten days after the digestive symptoms. In our study, hypoalbuminemia was observed in 85.7% of immunocompetent patients and 95.5% of immunosuppressed ones. However, it was significantly decreased in immunocompetent children and could explain the important proportion of oedema observed only in the immunocompetent patients. Indeed, the mild hypoalbuminemia level of immunosuppressed children with few oedema can be explained by their underlying disease, such as IBD [28], or by the early management of any new symptoms in this population (who easily visit emergency departments and have a close medical follow-up).

The immunocompetent children were also significantly younger (29.7 months) compared to the immunosuppressed ones (105.3 months), with a median age of CMV-associated PLE in the immunocompetent group rather matching with a primo-CMV-infection, whereas the older age in immunosuppressed patients could be due to reinfections and reactivations [2]. When performed, serologic tests were in line with this hypothesis. Already known risk factors of severe infection for CMV infection include being transplant recipients and immunosuppressed individuals [29], with primo-infections and reactivations both possibly life-threatening [30]. As expected, this group had a worse global prognosis with half of the immunosuppressed patients having a severe prognosis, including one patient’s death. Thus, early detection is important to avoid severe CMV disease in immunosuppressed patients [31], and CMV infections are commonly sought in this population. On the other hand, CMV-associated PLE is described as being benign and self-limited in immunocompetent children [12,13,14,24,26]. However, in our study, some immunocompetent children had an altered state, needed emergency care and five of them (23.8%) had a severe course. The median age for the children diagnosed with a severe course was 2.7 months [1.7–3.6], and they were all born at term and breast-fed. Four had colitis and one had gastritis with a systemic CMV disease. While severe infections are well documented in preterm infants, invasive CMV enterocolitis has been sporadically reported among term infants around 2.3 months old with transmission through breast milk [5,32]. Considering that CMV is the herpes virus with the highest morbidity and mortality rates and that a known risk factor for severe CMV infections is a very young age [29], CMV-associated PLE should hence be considered in healthy infants. Nevertheless, it should be noted that, in our study, the confirmed CMV-associated PLE case distribution is equivalent in both groups, underlying that this disease is not routinely sought in immunocompetent patients and probably underdiagnosed in less severe cases, with an overestimation of the prevalence of severe course.

The pathogenesis of CMV-associated PLE is not fully understood: CMV is assumed to induce the overexpression of transforming growth factor-α then to activate epidermal growth factor receptor signalling, stimulating foveolar mucous cell proliferation and foveolar hyperplasia and resulting in mucosal epithelium thickness. It also increases epithelium permeability through abnormal tight junctions, leading to a loss of serum proteins into the GI tract [14,33,34]. In immunocompetent patients, the biopsies with positive CMV PCR were consistent with the localisation of digestive symptoms for all patients suffering from colitis, but only for half of those who had gastritis. This is similar to the literature where the histological examination is useful in CMV colitis but not in CMV gastritis [12,35]. On the other hand, in immunosuppressed patients, the PCR positiveness on biopsy was correlated with their digestive signs for all localisations. However, to the best of our knowledge, no link has been established between the result of the EGD and the severity of the disease, except studies on CMV-associated protein-losing gastropathy that assumed that it follows a benign course [14,23]. In this study, on the EGD, the characteristics of CMV-associated PLE gastropathy are fundic gastritis and hypertrophic gastric fold [12,14,24], frequent in immunocompetent children but absent in immunosuppressed patients, which is not highlighted in the literature. The typical features of CMV were sparsely found. However, our results were not in accordance with the literature since, among the 17 patients with CMV-associated protein-losing-gastropathy, 56.3% had a benign prognosis, but two patients had a severe prognosis (12.5%) and the others a mild course. Conversely, among the 24 patients who had colitis, including 4 of the 5 immunocompetent patients with a severe prognosis, 12.5% had a benign prognosis and 58.3% had a severe one. Thus, while CMV colitis is known as a poor prognostic indicator among patients with IBD [36,37,38], in our study, it seems to have a poor prognosis in the overall paediatric population. However, the EGD results did not impact the treatment, with the macroscopic descriptions unreliable for the diagnosis, and no endoscopic follow-up of the digestive lesions was performed.

It is important to confirm the diagnosis using the most useful and the least invasive tests. Regardless of CMV PCR on digestive tissue, 88.9% of immunocompetent and 95.5% of immunosuppressed patients had evidence of CMV infection, based on CMV PCR on bodily fluids, except for one immunocompetent patient who had positive serology. Thus, although broader studies are needed to confirm these results, less invasive techniques such as CMV PCR on bodily fluid, which recently became the reference method [39], or serology should be favoured above endoscopy to confirm the diagnosis. Performing an endoscopy to diagnose CMV gastritis in healthy children, as is recommended in several studies [12,14,24], is questionable when this method has low adequacy, with poor typical CMV features and low-sensitivity CMV PCR on digestive tissue and induces no therapeutic change. Moreover, adverse events related to endoscopy cannot be overlooked, such as ventilatory problems, linked to sedation and anaesthesia, as well as perforation and bleeding, secondary to the endoscopic procedure [40,41,42]. Conversely, endoscopy should be encouraged to differentiate CMV-associated PLE and graft-versus-host disease in transplant patients [43,44]. Even though the histological findings are ambiguous, as they may be similar to a CMV infection (epithelial apoptosis, crypt abscesses, degeneration of crypt cells, nuclear atypia) [45], our study showed that the CMV PCR on digestive tissue was relevant for the diagnosis in immunosuppressed patients and will help differentiate both diagnoses in this specific indication.

This study included several limits due to its retrospective nature. Obviously, during the study period, children may have unconfirmed CMV-associated PLE, and such patients are probably more immunocompetent with a benign clinical course, leading to a selection bias. Moreover, in both hospitals, there is a paediatric oncology department, increasing the number of immunosuppressed patients with a diagnosis of PLE compared to the immunocompetent ones.

## 5. Conclusions

CMV-associated PLE should be foreseen in children and infants who present with oedema and hypoalbuminemia without abnormal urinary tests, especially if they have a history of digestive symptoms days before. Even if it is characterised by a benign course, some immunocompetent children, especially infants and patients with CMV colitis, could have an altered state and severe clinical course. Therefore, CMV infection should be confirmed using PCR analysis on bodily fluids and avoiding endoscopy, which is often unreliable for a prompt diagnosis, potentially leading to adverse effects and having no consequence on immediate medical management. Even in the immunosuppressed patients, EGD indications should be limited to organ and bone marrow transplant patients, to help differentiate a CMV infection from digestive graft-versus-host disease, through PCR CMV on digestive tissue more than the histological findings.

However, further prospective studies are needed to evaluate the prevalence of this disease in immunocompetent patients after including CMV research as a standard test when approaching a child with generalised oedema. Such studies will also highlight its risk factors and real severity in immunocompetent children.

## Figures and Tables

**Figure 1 nutrients-15-02844-f001:**
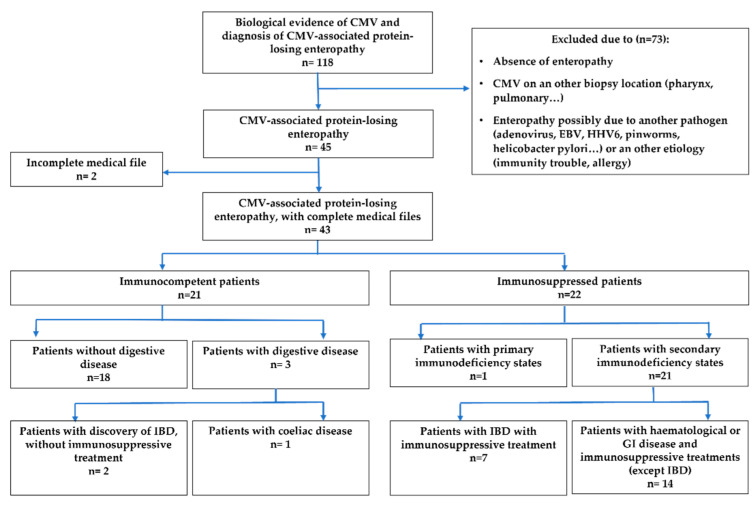
Flow chart.

**Table 1 nutrients-15-02844-t001:** Patients’ characteristics.

Clinicopathological Features	Immunocompetent Patients (*n* = 21)	Immunosuppressed Patients (*n* = 22)	Overall Population (*n* = 43)	Confidence Interval	*p*
Age in months, median [min-max]	29.7 [1–133.7]	105.3 [3.4–179.4]	45.9 [1–179.4]		*p* < 0.001
Age < 24 months, n (%)	9 (42.9)	2 (9)	11 (25.6)	[1.2; 78.8]	0.02
Gender, n (%)					
Male	13 (61.9)	14 (63.6)	27 (62.8)	[0.2; 3.8]	1
Female	8 (38)	8 (36.3)	16 (37.2)	[0.3; 4.4]	
Chronic digestive disease, n (%)					
Coeliac disease	1 (4.7)	0 (0)	1 (2.3)	[0.03; +∞[	0.5
Inflammatory bowel disease	2 (9.5)	7 (31.8)	9 (20.9)	[0.02; 1.5]	0.13
Immune deficiency, n (%)					
Primary immunodeficiency	/	1 (4.5)	1 (2.3)		1
Secondary immunodeficiency	/	21 (95.4)	21 (48.8)		*p* < 0.001

**Table 2 nutrients-15-02844-t002:** Clinical and biological signs of CMV-associated PLE in immunocompetent and immunosuppressed patients.

Clinical Symptoms	Immunocompetent Patients (*n* = 21)	Immunosuppressed Patients (*n* = 22)	Overall Population (*n* = 43)	Confidence Interval	*p*
Median time of onset of digestive symptoms before the first consultation, in days [interquartile range]	3 [1–6]	1 [0–4.8]	1 [0–5]		0.33
Oedema, n (%)	13 (61.9)	1 (4.5)	14 (32.6)	[3.6; 1502.4]	*p* < 0.001
Fever, low-grade fever, n (%)	10 (47.6)	12 (54.5)	22 (51.2)	[0.2; 2.9]	0.76
Vomiting, n (%)	18 (85.7)	11 (50)	29 (67.4)	[1.2; 39.2]	0.02
Abdominal pain, n (%)	11 (52.4)	11 (50)	22 (51.2)	[0.3; 4.3]	1
Acute diarrhoea, n (%)	14 (66.7)	19 (86.4)	33 (76.7)	[0.05; 1.7]	0.16
Dehydration, n (%)	9 (42.8)	6 (27.3)	15 (34.9)	[0.5; 8.8]	0.35
Gastrointestinal bleeding, n (%)	11 (52.4)	11 (50)	22 (51.2)	[0.3; 4.3]	1
Anorexia, n (%)	14 (66.7)	15 (68.2)	29 (67.4)	[0.2; 4]	1
Asthenia, n (%)	11 (52.4)	11 (50)	22 (51.2)	[0.3; 4.3]	1
**Signs of organ failure, n (%)**	**Immunocompetent patients (*n* = 21)**	**Immunosuppressed patients (*n* = 22)**	**Overall population (*n* = 43)**		** *p* **
Cardiac failure signs	0 (0)	0 (0)	0 (0)	/	1
Renal failure signs	0 (0)	0 (0)	0 (0)	/	1
Liver failure signs	0 (0)	1 (4.5)	1 (2.3)	/	1
**Biological signs**	**Immunocompetent patients (*n* = 21)**	**Immunosuppressed patients (*n* = 22)**	**Overall** **population** **(*n* = 43)**		** *p* **
Development of hypoalbuminemia at any time, n (%)	18 (85.7)	21 (95.5)	39 (90.7)	[0.005; 4]	0.34
Median duration of symptoms at the time of hypoalbuminemia, in days [interquartile range]	9.5 [6.3–18]	14 [2–18]	12 [5.5–18]		0.88
Median serum albumin, g/L [interquartile range]	21.2 [17.6–25.7]	29.6 [24.9–33.9]	25.4 [19.2–31.8]		0.01
Natremia ≤ 135 mmol/L, n (%)	12 (57.1)	5 (22.7)	17 (39.5)	[1; 21.4]	0.03
CRP > 10 mg/L, n (%)	7 (33.3)	15 (68.2)	22 (51.2)	[0.05; 1]	0.03

**Table 3 nutrients-15-02844-t003:** Identification of CMV in immunocompetent and immunosuppressed patients. Ratios represent the number of patients who had a positive test among those who had the test.

CMV Identification	Immunocompetent Patients (*n* = 21)	Immunosuppressed Patients (*n* = 22)	Overall Population (*n* = 43)	Confidence Interval	*p*
CMV PCR positive on bodily fluids, n (%)	13/16 (81.2)	21/22 (95.5)	34/38 (89.5)	[0.004; 3]	0.30
CMV PCR positive on digestive biopsy, n (%)	15/20 (75)	22/22 (100)	37/42 (88.1)	/	0.02
CMV serology					
Detection of CMV IgM antibodies, n (%)	9/10 (90)	2/5 (40)	11/15 (73.3)	[0.6; 778.9]	0.08
Detection of CMV IgG antibodies, n (%)	9/10 (90)	5/5 (100)	14/15 (93.3)	/	1
pp65 antigenemia, n (%)	1/1 (100)	3/4 (75)	4/5 (80)	[0.006; +∞[	1
**Biological evidence of CMV infection**					
Positive viral test including CMV PCR on digestive tissue, n (%)	21 (100)	22 (100)	43 (100)	/	1
Performed viral test excluding digestive CMV PCR, n (%)	18 (85.7)	22 (100)	40 (93)	/	0.1
Positive viral test excluding digestive CMV PCR, n (%)	16/18 (88.9)	21/22 (95.5)	37/40 (92.5)	[0.006; 8.1]	0.58

**Table 4 nutrients-15-02844-t004:** Treatments and disease course in immunocompetent and immunosuppressed patients.

Treatments	Immunocompetent Patients (*n* = 21)	Immunosuppressed Patients (*n* = 22)	Overall Population (*n* = 43)	Confidence Interval	*p*
In depth-treatment, n (%)					
Preventive antiviral drug	0 (0)	2 (9)	2 (4.7)	/	0.50
Corticosteroids	0 (0)	16 (72.3)	16 (37.2)	/	*p* < 0.001
Emergency treatment (<24 h), n (%)					
Fast vascular filing	4 (19)	3 (13.6)	7 (16.3)	[0.2; 11.6]	0.70
Intravenous or nasogastric rehydration	11 (52.4)	12 (54.5)	23 (53.5)	[0.2; 3.6]	1
Treatment during hospitalisation, n (%)					
Albumin infusion	8 (38)	7 (31.8)	15 (34.9)	[0.3; 5.6]	0.75
Curative antiviral drug	5 (23.8)	22 (100)	27 (62.8)	/	*p* < 0.001
Parenteral nutrition	5 (23.8)	10 (45.5)	15 (34.9)	[0.08; 1.6]	0.2
Blood transfusion	6 (28.6)	7 (31.8)	13 (30.2)	[0.2; 3.8]	1
Immunoglobulin infusion	2 (9.5)	7 (31.8)	9 (20.9)	[0.02; 1.5]	0.13
Intravenous corticosteroids	2 (9·5)	9 (40.9)	11 (25.6)	[0.01; 0.9]	0.03
**Clinical course, n (%)**	**Immunocompetent patients (*n* = 21)**	**Immunosuppressed patients (*n* = 22)**	**Overall** **population** **(*n* = 43)**		** *p* **
Global clinical course				/	0.02
Benign course	10 (47.6)	4 (18.2)	14 (32.6)	[0.9; 21.7]	0.05
Mild course	6 (28.6)	6 (27.3)	12 (27.9)	[0.2; 5]	1
Severe course	5 (23.8)	12 (54.5)	17 (39.5)	[0.06; 1.1]	0.06

## Data Availability

The data presented in this study are available on request from the corresponding author.

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
