# Peer review of "Clinical Manifestation of Cytomegalovirus-Associated Protein-Losing Enteropathy in Children"

_nutrients, 2023, doi:10.3390/nu15132844_

Round 1
Reviewer 1 Report
- The sample size is small, regardless being the largest cohort describing CMV-associated PLE in children. It would be worth for another study forming a consortium of multiple hospitals to gather more data.
- In "Thus, less invasive techniques such as CMV PCR on body fluid which recently became the reference method [49], or serology could have been favoured above endoscopy to confirm the diagnosis. " it is not necessarily true, I would rephrase it that would be for gastritis but not colitis. Recommending avoiding endoscopy with this small sample and results would not be advisable and should be mentioned that in the future it could be a possibility but not at the time since further studies need to be done.
- In the manuscript it is not mentioned follow up onto do a CMV sigmoidoscopy or colonoscopy topic and importance of such.
-I would recommend adding these articles to the manuscript:
"Dioverti MV, Razonable RR. Cytomegalovirus. Microbiol Spectr. 2016 Aug;4(4). doi: 10.1128/microbiolspec.DMIH2-0022-2015. PMID: 27726793."
Buck Q, Cho S, Mehta Walsh S, Schady D, Kellermayer R. Routine Histology-Based Diagnosis of CMV Colitis Was Rare in Pediatric Patients. J Pediatr Gastroenterol Nutr. 2022 Oct 1;75(4):462-465. doi: 10.1097/MPG.0000000000003528. Epub 2022 Jun 16. PMID: 35706089.
Kalkan IH, Dağli U. What is the most accurate method for the diagnosis of cytomegalovirus (CMV) enteritis or colitis? Turk J Gastroenterol. 2010 Mar;21(1):83-6. doi: 10.4318/tjg.2010.0061. PMID: 20549887.
Author Response
Dear reviewer,
On behalf of all the authors, I thank you for your comments and the recommended references, carefully read and added to the relevant places.
Concerning the small sample size, we are aware of this, and we are currently turning to a prospective collection to obtain a prevalence of the disease among children consulting with generalized oedema in pediatric emergencies as in the services directly receiving the immunocompromised children in several centres.
Concerning colitis and the interest in a follow-up of colonoscopies, no immunocompetent child had endoscopic follow-up and these examinations were very rarely performed again in immunosuppressed patients, only in cases of a long evolution, to eliminate an added cause. I specified it in the manuscript, as I qualified the sentence “Thus, less invasive techniques such as CMV PCR on body fluid which recently became the reference method [49], or serology could have been favoured above endoscopy to confirm the diagnosis.” However, we think that endoscopy has its place only when the doubt remains with digestive graft-versus-host disease in immunosuppressed patients and should be avoided.
I hope you will be satisfied with the changes made.
Reviewer 2 Report
The paper “Clinical manifestation of Cytomegalovirus-associated protein-losing enteropathy in Children” presents the results of the largest cohort describing CMV-associated protein-losing enteropathy in children. The topic is of interest and, even if of little novelty, the paper sheds light on an often underdiagnosed condition. The study is well-designed, and the results are clearly presented and extensively discussed. The main strength is the large sample size. I only suggest that five tables are probably redundant and table 4 could be moved in the supplementary material. Moreover, the critical message of thinking to CMV-associated PLE should be emphasized in the conclusions, addressing the red flags that could drive the clinician to consider that clinical entity, possibly without performing invasive testing that adds little or nothing to the correct management of such a condition. Finally, 55 references are probably too many.
Author Response
Dear reviewer,
On behalf of all the authors, I thank you for your comments.
Following your advice, we have placed Table 4 in the supplementary material, sought to reduce the number of references as much as possible, despite the ones added at the request of your co-reviewer, and the conclusion insists on the elements that should lead to a search for CMV-associated PLE, as the diagnostic approach.
I hope you will be satisfied with the changes made.